# Brain structures and activity during a working memory task associated with internet addiction tendency in young adults: A large sample study

**Saeid Sadeghi**[1,2,3]*, **Hikaru Takeuchi**[2], **Bita Shalani**[4], **Yasuyuki Taki**[2,5,6], **Rui Nouchi**[7,8,9], **Ryoichi Yokoyama**[10], **Yuka Kotozaki**[11], **Seishu Nakagawa**[12,13], **Atsushi Sekiguchi**[5,12,14], **Kunio Iizuka**[15], **Sugiko Hanawa**[12], **Tsuyoshi Araki**[16], **Carlos Makoto Miyauchi**[9], **Kohei Sakaki**[9], **Takayuki Nozawa**[17,18], **Shigeyuki Ikeda**[19], **Susumu Yokota**[20], **Daniele Magistro**[21], **Yuko Sassa**[2], **Ryuta Kawashima**[2,9,12]

1 Institute for Cognitive and Brain Sciences (ICBS), Shahid Beheshti University, Tehran, Iran, 2 Division of Developmental Cognitive Neuroscience, Institute of Development, Aging and Cancer, Tohoku University, Sendai, Japan, 3 Center of Excellence in Cognitive Neuropsychology, Shahid Beheshti University, Tehran, Iran, 4 Department of Psychology, Faculty of Humanities, Tarbiat Modares University, Tehran, Iran, 5 Division of Medical Neuroimaging Analysis, Department of Community Medical Supports, Tohoku Medical Megabank Organization, Tohoku University, Sendai, Japan, 6 Department of Radiology and Nuclear Medicine, Institute of Development, Aging and Cancer, Tohoku University, Sendai, Japan, 7 Creative Interdisciplinary Research Division, Frontier Research Institute for Interdisciplinary Science, Tohoku University, Sendai, Japan, 8 Human and Social Response Research Division, International Research Institute of Disaster Science, Tohoku University, Sendai, Japan, 9 Department of Advanced Brain Science, Institute of Development, Aging and Cancer, Tohoku University, Sendai, Japan, 10 School of Medicine, Kobe University, Kobe, Japan, 11 Division of Clinical research, Medical-Industry Translational Research Center, Fukushima Medical University School of Medicine, Fukushima, Japan, 12 Department of Human Brain Science, Institute of Development, Aging and Cancer, Tohoku University, Sendai, Japan, 13 Department of Psychiatry, Tohoku Medical and Pharmaceutical University, Sendai, Japan, 14 Department of Behavioral Medicine, National Institute of Mental Health, National Center of Neurology and Psychiatry, Tokyo, Japan, 15 Department of Psychiatry, Tohoku University Graduate School of Medicine, Sendai, Japan, 16 ADVANTAGE Risk Management Co., Ltd., Tokyo, Japan, 17 Research Center for the Earth Inclusive Sensing Empathizing with Silent Voices, Tokyo Institute of Technology, Tokyo, Japan, 18 Graduate School of Arts and Sciences, Department of General Systems Studies, The University of Tokyo, Tokyo, Japan, 19 Department of Ubiquitous Sensing, Institute of Development, Aging and Cancer, Tohoku University, Sendai, Japan, 20 Division for Experimental Natural Science, Faculty of Arts and Science, Kyushu University, Fukuoka, Japan, 21 Department of Sport Science, School of Science and Technology, Nottingham Trent University, Nottingham, United Kingdom

* sae_sadeghi@sbu.ac.ir

**Data Availability Statement:** All the experimental data obtained in the experiment of this study will be available to ones that were admitted in the ethics

## Abstract

The structural and functional brain characteristics associated with the excessive use of the internet have attracted substantial research attention in the past decade. In current study, we used voxel-based morphometry (VBM) and multiple regression analysis to assess the relationship between internet addiction tendency (IAT) score and regional gray and white matter volumes (rGMVs and rWMVs) and brain activity during a WM task in a large sample of healthy young adults (n = 1,154, mean age, 20.71 ± 1.78 years). We found a significant positive correlation between IAT score and gray matter volume (GMV) of right supramarginal gyrus (rSMG) and significant negative correlations with white matter volume (WMV) of right temporal lobe (sub-gyral and superior temporal gyrus), right sublobar area (extra-

committee of Tohoku University, school of medicine. All the data sharing should be first admitted by the ethics committee of Tohoku University, school of medicine. The contact information of the ethics committee is as follows (* should be replaced by @). med-kenkyo*grp. tohoku.ac.jp.

**Funding:** This study was supported by a Grant-in-Aid for Young Scientists (B) (KAKENHI 23700306) and a Grant-in-Aid for Young Scientists (A) (KAKENHI 25700012) from the Ministry of Education, Culture, Sports, Science, and Technology. The funder provided support in the form of salaries for authors H.T., but did not have any additional role in the study design, data collection and, analysis, decision to publish, or manuscript preparation of the manuscript. The specific roles of these authors are articulated in the '"author contributions' contributions" section.

**Competing interests:** The authors declare no competing financial or non-financial interests.

nuclear and lentiform nucleus), right cerebellar anterior lobe, cerebellar tonsil, right frontal lobe (inferior frontal gyrus and sub-gyral areas), and the pons. Also, IAT was significantly and positively correlated with brain activity in the default-mode network (DMN), medial frontal gyrus, medial part of the superior frontal gyrus, and anterior cingulate cortex during a 2-back working memory (WM) task. Moreover, whole-brain analyses of rGMV showed significant effects of interaction between sex and the IAT scores in the area spreading around the left anterior insula and left lentiform. This interaction was moderated by positive correlation in women. These results indicate that IAT is associated with (a) increased GMV in rSMG, which is involved in phonological processing, (b) decreased WMV in areas of frontal, sublobar, and temporal lobes, which are involved in response inhibition, and (c) reduced task-induced deactivation of the DMN, indicative of altered attentional allocation.

## Introduction

The internet is a necessity in many lives [1]. More than half of the world's population are internet users [2, 3]. Excessive internet use is associated with negative psychological consequences such as poor life satisfaction [4, 5], anxiety and aggression [6, 7], low self-esteem and depression [8, 9], and alcohol abuse [10, 11]. Physical health problems such as sleep problems [12–14] and social functioning impairments such as poor academic performance [15, 16] are other negative consequences of excessive internet use.

Excessive internet use has also been associated with impaired executive functions [17–21]. In addition, some studies have also indicated that internet users show working memory (WM) deficits compare to individuals without such behaviors [19–22].

WM is a central component of executive functioning [23]. It has been suggested that WM along with inhibition and shifting contribute to self-regulation [24]. Prior research suggests that WM is a significant predictor of the ability to have proper response inhibition [25]. WM deficits have been observed in individuals with hyperactivity and attention disorder and impulsivity [26, 27], substance-dependent individuals, including cocaine- [28], alcohol- [29], methamphetamine- [30] and opioid-dependent individuals [31]. WM load interferes with individuals' ability to filter out irrelevant distractors [32]. Also, there is evidence of significant conjunction between WM and response inhibition in the left inferior frontal gyrus [33].

During cognitive tasks performance, the default-mode network (DMN) is deactivated [34]. DMN is a set of brain regions (posterior cingulate/precuneus, medial prefrontal cortex) considered a backbone of cortical integration [35–38]. Previous studies have revealed that task-induced deactivations occur within regions of the DMN during cognitive WM tasks [39–43]. In addition, a reduced magnitude of task-induced deactivation in the DMN is a characteristic of subjects with lower WM capacity and cognitive disinhibition [44, 45]. Global Workspace Theory [46] has helped researchers understand how WM relates to the DMN. In brief, the theory postulates that the central executive (CE) component of the WM model presides over cognitive slave systems to orchestrate conscious cognitive control of distracting stimuli. The CE is related to the executive control network and functions antagonistically to the DMN.

However, the relationship between characteristics of brain activity during a WM task and the tendency of people to internet addiction (IA) has not been studied yet. One of the aims of this study is to understand the characteristics of brain activity during a WM task associated with IAT.

Research has focused on internet addiction disorder (IAD) in pathological groups rather than IAT in healthy people groups. Magnetic resonance imaging (MRI) studies have revealed that internet addiction (IA) scores negatively correlate with GMVs in the anterior cingulate cortex (ACC), bilateral dorsolateral prefrontal cortex (DLPFC), orbitofrontal cortex (OFC), right middle frontal gyrus, supplementary motor area (SMA), cerebellum, left rostral anterior cingulate cortex (rACC), and post-central gyrus (postCG) [47–49]. Lin, Zhou [50] also used diffusion tensor imaging (DTI) to investigate white matter integrity in adolescents with IAD. This study reported that people with higher IAD scores appeared to have lower white matter integrity in the fronto-temporal pathway connected through the external capsule. Takeuchi, Taki [51] have shown that video game time is associated with increased mean diffusivity (MD) in the orbital frontal cortex and subcortical areas (putamen, pallidum, left hippocampus, caudate, right insula, and thalamus). Takeuchi, Taki [52] also demonstrated in a longitudinal study that excessive internet use is associated with decreased verbal intelligence and a smaller developmental increase in rGMVs and rWMVs, respectively across widespread brain areas in children.

Moreover, functional magnetic resonance imaging (fMRI) studies have shown that the most cortical dysfunctions in IAD are reported to be localized to the superior temporal gyrus [53], cingulate cortex [54], cerebellum [55], and inferior frontal gyrus. In subcortical regions, functional alterations were often found in the brainstem and caudate [56]. Previous task-related fMRI studies on IAD have demonstrated differences in behavioral performance and differences in brain activation during cognitive tasks such as cue-reactivity paradigms in which subjects are exposed to internet or videogame stimuli to elicit a craving, probabilistic guessing paradigms in which subjects bet using cards or colors to analyze neural reward system dynamics in response to losses or wins, and cognitive control paradigms such as the GO-NO-GO test for assessment of impulsivity and inhibitory control [56].

Although such combined behavioral and neuroimaging studies have shown that IAD is associated with altered brain structure [56], but due to small sample sizes [57] and diversity in empirical research methods and paradigms in neuroimaging studies [58] results are inconsistent and often are not replicated. Also, previous studies have all focused on the group of people with IAD, and the study of IAT in healthy people has been neglected. With a large sample size, the current study focuses on the tendency of IA in healthy people to increase our knowledge about the nature of the phenomenon of IA. For these reasons, future studies are warranted.

The purpose of this study was thus to investigate these issues by assessing the effects of IAT on brain structure and activity during the n-back working memory task in a large sample of healthy young adults. Knowledge of the brain structure and function abnormalities and association between these abnormalities and IAT is helpful to identify possible interventions and pharmacotherapies to treat IA.

On the basis of the previous studies, we hypothesized that higher IAT scores may be associated with structural abnormalities in the frontal and temporal lobe and subcortical areas known to contribute to addiction vulnerability [50, 59–61]. We also hypothesized that lower task-induced deactivation (TID) in the DMN during WM may be associated with a higher IAT score. This hypothesis is based on previous findings that suggest that TID in the DMN is associated with altered brain glutamatergic excitability and gamma aminobutyric acid (GABA) inhibition [62], that the glutamatergic neurons play a critical role in the reward system [63], and that glutamatergic and GABAergic abnormalities are primary neurobiological characteristics in individuals with addiction [64–66]. We also hypothesized that lower TID in the DMN during WM may be associated with a higher IAT score, which is supported by previous studies that showed reduced task-induced deactivation in the DMN during working memory tasks in psychiatric patients [67–69].

## Materials and methods

### Participants

A total of 1,154 healthy right-handed young adults (666 men, mean age 20.79 years, standard deviation = 1.89 years and 488 women, mean age 20.60 years, standard deviation = 1.61 years) participated in this study as part of our ongoing project to explore the associations among brain imaging characteristics, cognitive functions, aging, genetics, and daily habits. Indeed, from our database, we used the data from 1,154 subjects that had questionnaire data about internet dependence, fMRI imaging data, and behavioral data of the N-back task without apparent artifacts.

All subjects were students from Tohoku University and neighboring universities and colleges in Japan. All but one of the subjects in this study were native Japanese speakers. However, one foreign Asian student who was very proficient in Japanese and was determined to be equipped to go through the experimental Japanese procedures like native Japanese speakers was allowed to participate in this study. The removal of this one subject affects the results little. They were recruited using advertisements on bulletin boards at Tohoku University or via e-mail introducing the study. These advertisements and e-mails specified the exclusionary characteristics in individuals regarding participation in the study, such as handedness, the existence of metal in and around the body, claustrophobia, the use of certain drugs, and a history of certain psychiatric disorders and neurological diseases, and previous participation in related experiments. A history of psychiatric and neurological diseases and/or recent drug use was assessed using our laboratory's routine questionnaire, in which each subject answered questions related to their current or previous experiences of any of the listed diseases and listed drugs that they had recently taken. The questionnaire also asked the personal contact information, age, birthday, the institutes they belong to, age, sex, weight, and height. The Edinburgh Handedness Inventory [70] was also included in this routine questionnaire. We used the Edinburgh Handedness Inventory to evaluate handedness in subjects. Previous studies demonstrated significant differences in brain morphology and activity patterns between right-handers and left-handers [71–75]. For this reason, fMRI studies tend to exclude left-handers.

The scans were checked for noticeable brain lesions and tumors in this experiment, but no subjects had such apparent lesions or tumors. These descriptions are mostly obtained from our previously published work [76]. The participant's socio-demographic characteristics are presented in Table 1. The Ethics Committee of Tohoku University approved all procedures, which were performed in accordance with relevant guidelines and regulations. Written informed consent was obtained from each subject for the projects in which they participated. Descriptions in this subsection are adapted from a previous study using similar methods [77].

### Internet addiction tendency assessment

We used the Japanese version of Young's IAT scale to assess condition severity [78]. This IAT instrument consists of 20 items answered on a 1–5 scale from 1 = rarely to 5 = always. The scale is self-administered and requires 5 to 10 minutes. IAT measures the impact of internet use on people's daily routine, social life, productivity, sleeping pattern, and feelings. The IAT scale minimum and maximum scores are 20 and 100, with higher scores reflecting a greater tendency toward internet addiction. The developer of this scale suggests that a score of 20–39 points is an average online user who has complete control over his/her usage; a score of 40–69 signifies frequent problems due to internet usage, and a score of 70–100 means that the internet is causing significant problems [78]. The Japanese version of this scale has demonstrated high reliability and validity [79].

**Table 1. The socio-demographic characteristics of participants.**

| Variable | *Min* | *Max* | *M* | *SD* |
|---|---|---|---|---|
| Age (year) | 18 | 27 | 20.71 | 1.78 |
| Self-reported height | 142 | 192 | 166.35 | 8.44 |
| Self-reported weight | 38 | 115 | 57.96 | 9.50 |
| BMI | 15.39 | 32.88 | 20.83 | 1.74 |
| Family annual income [a] | 1 | 7 | 4.19 | 1.56 |
| parent years of education [b] | 9 | 21 | 14.69 | 1.85 |

[a] Family annual income was classified as follows: 1, annual income below 2 million yen; 2, 2–4 million yen; 3, 4–6 million yen; 4, 6–8 million yen; 5, 8–10 million yen; 6, 10–12 million yen; 7, >12 million yen; the currency exchange rate is approximately $1 USD = 120 yen.

[b] Parent average educational qualification (years of education) was classified as follows: 6 years, elementary school graduate or below; 9 years, junior high school graduate; 11 years, graduate of a short-term school completed after junior high school; 12 years, normal high school graduate; 14 years, graduate of a short-term school completed after high school (such as a junior college); 16 years, university graduate; 18 years, Master's degree; and 21 years, doctorate.

## fMRI task

Functional MRI was used to map brain activity during working memory. The n-back task is a widely used task consisting of 0-back (simple cognitive processing) and 2-back (working memory) conditions. In the 2-back task, subjects viewed a series of stimuli presented sequentially (one of four Japanese vowels) and were instructed to judge if a target stimulus appearing 2 presentations earlier was the same as the current stimulus by pushing a button. In the 0-back task, subjects were instructed to determine whether a presented letter was the same as the target stimulus by pushing a button (Fig 1). We used a simple block design. Descriptions in this subsection were mostly adapted from our previous studies using similar methods [77, 80].

In this study, our focus was TID in the DMN. TID in the DMN occurs in mostly similar areas regardless of whether the task is 2-back or 0-back, although there are differing magnitudes. Furthermore, differences in brain activity between patients with schizophrenia and control subjects were similar regardless of whether the task was a 0-back task or 2-back. These included areas of DMN (i.e., subtracting the activity during the 0-back task from the brain activity during the 2-back task substantially eliminates group differences) [44, 81]. Therefore, we did not analyze the contrast of 2-back– 0-back, as was done in another study that focused on TID in the DMN [44].

## Image acquisition

The MRI acquisition methods were described in our previous study [82]. Briefly, all MRI data were acquired using a 3 Tesla (3T) Philips Achieva scanner. Diffusion-weighted data were acquired using a spin-echo Echo planar imaging (EPI) sequence [repetition time (TR) = 10293 millisecond (ms), echo time (TE) = 55 ms, field-of-view (FOV) = 22.4 centimeter (cm), 2×2×2 millimeter (mm)$^3$ voxels, 60 slices, sensitivity encoding (SENSE) reduction factor = 2, number of acquisitions = 1]. The diffusion weighting was isotropically distributed along 32 directions ($b$ value = 1,000 s/mm$^2$). In addition, three images with no diffusion weighting ($b$ value = 0 s/mm$^2$ or b = 0 images) were acquired using a spin-echo EPI sequence (TR = 10293 ms, TE = 55 ms, FOV = 22.4 cm, 2 × 2 × 2 mm$^3$ voxels, 60 slices). For the n-back session, 174 functional volumes were obtained [77]. High-resolution T1-weighted structural images were collected

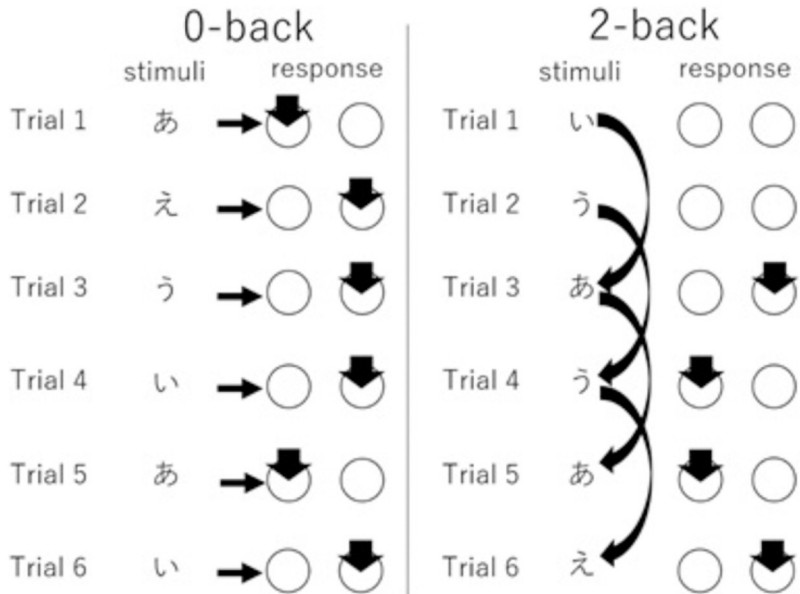

**Fig 1. A schematic diagram of the procedures used for the N-back task.**

using a magnetization-prepared rapid gradient echo sequence (T1WIs: 240 × 240 matrix, TR = 6.5 ms, TE = 3 ms, FOV = 24 cm, slices = 162, slice thickness = 1.0 mm).

## Preprocessing of structural data

Preprocessing of the structural and functional data was performed using Statistical Parametric Mapping (SPM) software (SPM12; Wellcome Department of Cognitive Neurology, London, UK) implemented in MATLAB (Mathworks, Inc., Natick, MA). For analyses, T1-weighted structural images of each individual were segmented using the new segmentation algorithm implemented in SPM12 and normalized to Montreal Neurological Institute (MNI) space to yield images with $1.5 \times 1.5 \times 1.5$ mm$^3$ voxels using the diffeomorphic anatomical registration through exponentiated lie algebra registration process implemented in SPM12. In addition, we performed a volume change correction (modulation) [83]. Subsequently, generated rGMV and rWMV images were smoothed by convolution using an isotropic Gaussian kernel of 8 mm full width at half maximum. These descriptions were mostly adapted from our previous study using similar methods.

## Pre-processing and data analysis for functional activation data

Pre-processing and data analysis of functional activation data were performed using SPM. The following procedures for functional activation data analysis were reproduced from our previous study, as described previously [84]. From the images collected, fractional anisotropy (FA) and mean diffusion (MD) maps were calculated [85]. In current study, these FA and MD maps were used during preprocessing of BOLD images. Prior to analysis, individual BOLD images were re-aligned and resliced to the mean BOLD image and then corrected for slice timing. Also, the abovementioned mean BOLD image was then realigned to the mean b = 0 image as previously described together with slice timing corrected images [77]. As the mean b = 0 image was aligned with the FA image and MD map, the BOLD image, b = 0 image, FA image, and MD map were all aligned.

All images were subsequently normalized using a previously validated two-step "new segmentation" algorithm of diffusion images and a previously validated diffeomorphic anatomical registration through exponentiated lie algebra (DARTEL)-based registration process [86]. This normalization method was used for all diffusion images, including gray matter segments (regional gray matter density [rGMD] map), white matter segments (regional white matter density [rWMD] map), and cerebrospinal fluid (CSF) segments (regional CSF density [rCSFD] map). Using the newly implemented segmentation algorithm in SPM8, the FA images of each individual were segmented into six tissues (first new segment). In this process, default parameters and tissue probability maps were used, except that affine regularization was performed using the International Consortium for Brain Mapping (ICBM) template for East Asian brains, and the sampling distance (the approximate distance between sampled points when estimating the model parameters) was 2 mm.

Next, we synthesize the FA image and the MD map. In this synthesized image, the area with WM tissue probability > 0.5 in the aforementioned new segmentation process was the FA image multiplied by −1. Hence, this synthesized image shows a very clear contrast between WM and other tissues. The remaining area is the MD map.

We continued with the DARTEL registration process implemented in SPM8. During this process, we used the DARTEL import image of the GM tissue probability map produced by the second new segmentation process as the GM input for the DARTEL process. First, the raw FA image was multiplied by the WM tissue probability map from the second new segmentation process within the areas with a WM probability > 0.5 (the signals from all other areas were set to 0). Then, this FA image×the WM tissue probability map was coregistered and resliced to the DARTEL import WM tissue probability image from the second segmentation. The template for the DARTEL procedures was generated using imaging data from the 63 subjects who participated in [77] and in the present study. Then, using this existing template, DARTEL procedures were conducted. The parameters have been changed as follows to improve the accuracy of the procedures. The number of Gauss–Newton iterations to be performed within each outer iteration was set to 10. In each outer iteration, we used 8-fold more timepoints than the default values to solve the partial differential equations. The number of cycles used by the full multi-grid matrix solver was set to 8. The number of relaxation iterations performed in each multi-grid cycle was also set to 8. The resultant synthesized images were spatially normalized to Montreal Neurological Institute (MNI) space. The voxel size of the normalized BOLD image is 3 3 3 mm3.

A design matrix was fitted to each participant with one regressor for each task condition (0-, 2-back in the n-back task) using the standard hemodynamic response function. The design matrix weighted each raw image according to its overall variability, to reduce the impact of movement artifacts [87]. The design matrix was fit to the data for each participant individually. After estimation, beta images were smoothed (8 mm full width half maximum) and taken to the second-level or subjected to a random effect analysis. We removed low-frequency fluctuations using a high-pass filter and a cutoff value of 128 s. The individual-level statistical analyses were performed using a general linear model.

In the individual analyses, we focused on activation related to the condition (0-back or 2-back versus rest). The resulting maps representing brain activity during the working memory condition (2-back) and simple cognitive processing condition (0-back) for each participant were then forwarded for group analysis.

The fMRI images with artifacts based on the visual inspection had been removed from the images. Thorough instruction to prevent motion during the scan was given to educated participants. Other exclusions based on motion parameters were not performed in this study.

In a previous study, we validated normalization procedures of fMRI using diffusion tensor images using SPM 8 [86]. Our internal preliminary survey also showed these procedures work better using SPM8. Conversely, VBM procedures work better with SPM12. In other words, the segmentation of the diffusion images obtained, which were part of our preprocessing procedures of fMRI, were not adequate for SPM12. Misclassifications that were apparent by visual inspection were systematically found when SPM 12 was used. In the second-level analysis, the use of SPM8 or SPM12 does not affect the results of threshold-free cluster enhancement (TFCE) based on permutation.

Generally, thorough instructions and thorough fixation by the pad were provided to prevent head motion during the scan as much as possible, and we utilized the software to reduce the impact of movements [87], as described in the subsection below.

Thus, we did not exclude any subject from the fMRI analyses based on excessive motion that did not cause evident artifacts during the scan. The subjects were young adults and the scan duration was very brief. Only the maximum movement of several subjects detracted from the original point, and in one of the directions exceeded 3 mm. Removing these subjects from analyses did not substantially alter the significant results of the present study.

Similarly, the subjects enrolled in the study were educated young adults, and thorough instruction and sufficient practice was provided. Subjects whose responses were properly recorded showed acceptable accuracies and only seven subjects showed accuracies lower than 80% in the 0-back or 2-back task (but accuracies were at least 50% or greater). Removing these subjects also did not substantially alter the significant results of the present study.

## Effects of interaction between sex and the score of Young's IAT scale on imaging measures

We also performed a supplementary investigation of the potential regions displaying significant effects of interaction between the subject's sex and score on the Young's IAT scale (that is, we investigated whether some regions showed sex-related differences in the correlations patterns based on the Young's IAT scale score). For this purpose, we performed whole-brain analyses of covariance (ANCOVAs). The dependent variables in these analyses were same as those in the whole-brain multiple regression analyses that were conducted to investigate the correlation with score of Young's IAT scale in each voxel across sexes. In these whole-brain ANCOVAs, sex was a group factor (using the full factorial option in SPM8), whereas and all other covariates are same as those of the abovementioned whole-brain multiple regression analyses. In addition, all covariates were modeled to enable unique relationships with imaging measures (dependent variables) (using the interactions option in SPM8) for each sex. The interaction between sex and scores on Young's IAT scale were assessed using t-contrasts. Correction for multiple comparisons was performed using the same method used in the whole-brain multiple regression analyses.

## Supplemental methods

**Supplemental analyses of the comparison between subjects using Young's IAT scale.** In accordance with the considerable literature available that classified subjects based on the Young's IAT scale score [88], we also divided subjects into two groups (IAT score $\geq$50 and IAT score <50). This classification was used to compare dependent variables between those who used the internet excessively and those who used it less frequently. We hypothesized that excessive use of the internet would be associated with additional changes in brain structure and functional characteristics. For this reason, we also conducted the supplemental analyses of

comparisons between subjects who scored ≥50 using Young's IAT scale and those who scored <50 using Young's IAT scale, on the basis of the criteria described previously [78].

For this comparison, we conducted multiple regression analyses in which all dependent and independent variables of the main analyses remained the same, except that Young's IAT score was replaced by the dichotomized value (Young's IAT scale ≥50 = 1, Young's IAT scale <50 = 0).

**Supplemental region of interest (ROI) analyses of the associations between activity in key nodes of the DMN and IAT scores.** We conducted a supplemental partial correlation analyses of the associations between mean beta estimates of functional ROIs of important nodes of the DMN and IAT after controlling for covariates. In these analyses, ROI masks were defined by the areas that are mostly significantly deactivated during the 2-back task using an appropriate T score threshold that successfully segregated each area in the representative DMN nodes for the 63 subjects from which the template of normalization was created (when there were multiple clusters in one area, those that showed the strongest statistical values at the peak were selected). The mean beta estimates of the 2-back task as well as the 0-back task within each ROI were extracted. For these analyses, control variables were same as those of the covariates in the whole-brain multiple regression analyses in the main text.

ROIs were medial prefrontal cortex (mPFC) (peak coordinate: x = −6, y = 57, z = −6, T score threshold = 15, 425 voxels), posterior cingulate cortex (PCC)/precuneus (peak coordinate: x = −6, y = −57, z = 12, T score threshold = 15, 305 voxels), left hippocampus (peak coordinate: x = −27, y = −21, z = −24, T score threshold = 9, 27 voxels), right hippocampus (peak coordinate: x = 24, y = −15, z = −27, T score threshold = 9, 66 voxels), left temporoparietal junction (peak coordinate: x = −45, y = −72, z = 21, T score threshold = 7, 322 voxels), and right temporoparietal junction (peak coordinate: x = 54, y = −69, z = 27, T score threshold = 7, 32 voxels).

Results with a threshold having $p < 0.05$, and corrected for the false discovery rate (FDR) using the two-stage sharpened method [89], were considered statistically significant.

### Statistical analysis

Statistical analyses of imaging data were performed with SPM8. Structural whole-brain multiple regression analyses were performed to investigate associations of IAT scores with rGMV and rWMV. Age, sex, and total intracranial volume calculated using voxel-based morphometry (for details of calculation see [90]) were added as covariates.

For the functional images, we used multiple regression analysis to investigate the relationship between IAT score and brain activity levels during the 0-back, 2-back, and 2-back-0-back tasks. Age, sex, n-back task accuracy, and n-back task reaction time were entered into the multiple regression model as covariates.

A multiple comparison correction was performed using TFCE [91] with randomized (5,000 permutations) nonparametric testing using the TFCE toolbox (http://dbm.neuro.uni-jena.de/tfce/). We applied a threshold of family-wise error corrected at P < .05. SPM8 was used for analyses because of better compatibility with TFCE software and our in-house scripts [52].

## Results

### Behavioral results

There was no significant difference in mean age between sexes, but independent sample t-tests revealed a significant difference in the IAT score. Moreover, there was no significant difference in 2-back accuracy and reaction time between sexes. The distribution of IAT scores by sex is presented in Fig 2, and the t-test results are presented in Table 2.

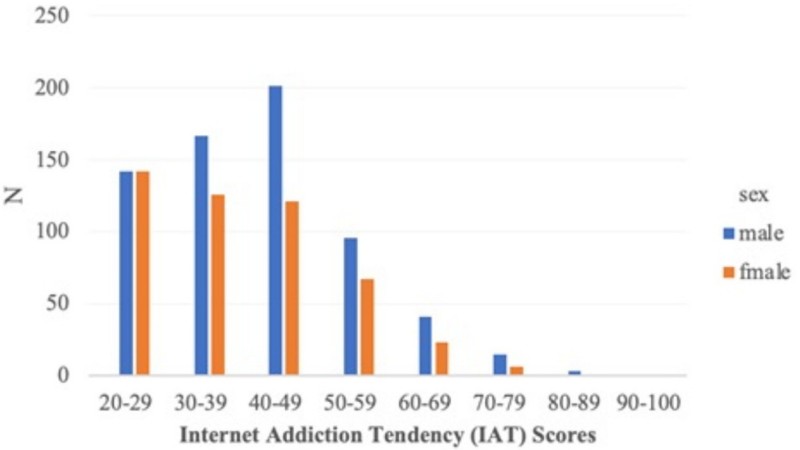

**Fig 2. Distribution of internet addiction tendency (IAT) scores.**

Furthermore, to investigate the relationship between IA and 2-back accuracy and 2-back reaction time, Pearson's correlations were conducted. There were no significant correlations between IA and 2-back accuracy ($r = 0.029$, $p = 0.33$) and 2-back reaction time ($r = -0.042$, $p = 0.15$).

## Structural results

VBM revealed a significantly correlation between IAT score and rGMV of rSMG among the entire cohort (Table 3 and Fig 3) as well as significant negative correlations between IAT score and rWMVs of right temporal lobe (sub-gyral and superior temporal gyrus), right sublobar region (extra-nuclear and lentiform nucleus), right cerebellum anterior lobe, cerebellar tonsil, right frontal lobe (inferior frontal gyrus and sub-gyral), and pons (Table 4 and Fig 4).

## fMRI results

Multiple regression analysis revealed that IAT scores were significantly and positively correlated with brain activity during the 2-back task in the medial frontal gyrus, superior frontal gyrus, and medial part of the ACC (Table 5 and Fig 5). This cluster of significant correlation mostly belonged to areas that were deactivated during the 2-back task (Table 5).

**Table 2. Comparison of IAT scores between men and women.**

| Variable | | Sex | M | SD | MD | Df | t | p |
|---|---|---|---|---|---|---|---|---|
| Age | | Man | 20.79 | 1.89 | 0.19 | 1152 | 1.87 | 0.062 |
| | | Woman | 20.60 | 1.61 | | | | |
| Internet addiction tendency | | Man | 41.32 | 13.10 | 2.70 | 1152 | 3.52 | 0.0001 |
| | | Woman | 38.62 | 12.58 | | | | |
| Working Memory | 2-back accuracy | Man | 0.99 | 0.030 | -0.11 | 1152 | -1.15 | 0.25 |
| | | Woman | 1.10 | 2.36 | | | | |
| | 2-back reaction time (sec) | Man | 0.6688 | 1769.27 | -41.78 | 1152 | -0.387 | 0.698 |
| | | Woman | 0.6729 | 1862.29 | | | | |

**Abbreviations**: M, mean; SD, standard deviation; MD, mean differences; df, degree of freedom; sec, second.

**Table 3. Brain gray matter regions with a significant positive main effect of IAT score on volume.**

| Anatomical area | MNI coordinates | | | TFCE value | Corrected *p* value (FWE) | Cluster size (mm³) |
|---|---|---|---|---|---|---|
| | X | Y | z | | | |
| Right supramarginal gyrus | 63 | -23 | 47 | 1193.41 | 0.044 | 250 |

Abbreviations: GM, gray matter: L, left: R, right; MNI, Montreal Neurological Institute; TFCE, threshold-free cluster enhancement.

**Effects of interaction between sex and IAT score.** There were only significant effects in the interaction between sex and the score of Young's IAT scale in the whole-brain analyses of rGMV. The significant effects of interaction were found in the area around the left anterior insula and left lentiform nucleus ($p = 0.021$, corrected, x, y, z = −24, 16.5, 9, TFCE score 1402.35, 3791 mm³ under the threshold of $p < 0.05$, corrected) (Fig 6). This interaction had a positive correlation in women ($r = 0.1822$, $p < .001$) and no correlations in men ($r = −0.054$, $p = 0.163$).

## Supplemental results

**Supplemental analyses of the comparison between subjects using Young's IAT scale.** The rGMV analysis revealed there were no significant correlations between rGMV and the dichotomized value (Young's IAT scale ≥50 = 1, Young's IAT scale < 0). However, this analysis revealed that subjects who scored ≥50 using Young's IAT scale had a tendency of greater rGMV in a similar area of significant correlation between rGMV and Young's IAT scale (right Inferior parietal lobule, x, y, z = 42, −49.5, 51, 1166 mm³ under the threshold of $p < 0.001$, uncorrected).

The rWMV analysis revealed that subjects who scored ≥50 using Young's IAT scale had lower rWMV, with significant negative correlations between rWMV and the dichotomized value (Young's IAT scale ≥50 = 1, Young's IAT scale < 0), in the left frontal white matter area ($p = 0.026$, corrected, x, y, z = −21, 46.5, 6, 7395 mm³), in the right frontal white matter area ($p = 0.036$, corrected, x, y, z = 31.5, 25.5, −1.5, 2333 mm³), and in the white matter area in the cerebellum and the brain stem ($p = 0.040$, corrected, x, y, z = 19.5, −24, −30, 2749 mm³). These

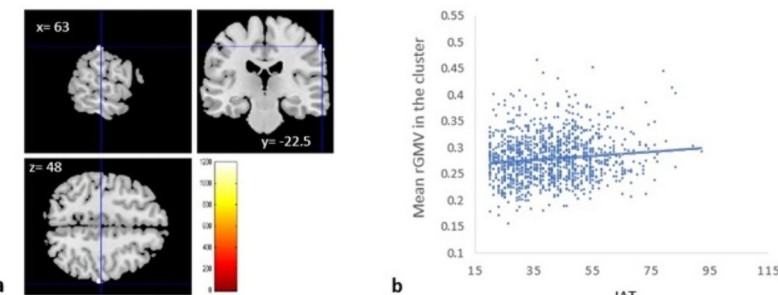

**Fig 3. Regional gray matter volumes correlated with internet addiction tendency (IAT) score in young adults.** (a) The panels show the areas of significant positive correlation between IAT score and rGMV. The results shown were obtained using a threshold of threshold-free cluster enhancement (TFCE) of $p < 0.05$ based on 5,000 permutations. A significant positive correlation was found in the right supramarginal gyrus. (b) Scatterplot of the association between IAT score and mean rGMV values of the significant cluster. IAT is positively correlated with mean rGMV of the significant cluster in men (r = 0.10, $p = 0.01$), and in women (r = 0.099, $p = 0.029$).

**Table 4. Brain white matter regions with a significant negative main effect of IAT score on volume.**

| Cluster | Lobe (L/R) | Nearest WM area | MNI coordinates | | | TFCE value | Corrected $p$ value (FWE) | Cluster size (mm³) |
|---|---|---|---|---|---|---|---|---|
| | | | x | Y | Z | | | |
| 1 | Temporal (R) | Sub-Gyral | 23 | -53 | 15 | 1742.07 | 0.007 | 113825 |
| | Sublobar (R) | Extra-Nuclear | 24 | -39 | 14 | 1635.53 | 0.008 | |
| | Temporal (R) | Superior temporal gyrus | 42 | -35 | 6 | 1615.76 | 0.008 | |
| 2 | Cerebellum posterior (R) | Cerebellar tonsil | 14 | -47 | -44 | 1621.78 | 0.008 | 42741 |
| | Brain stem (R) | Pons | 18 | -35 | -33 | 1618.07 | 0.008 | |
| | Cerebellum anterior (R) | cerebellum anterior lobe | 21 | -44 | -39 | 1595.88 | 0.008 | |
| 3 | Frontal (R) | Sub-gyral | 27 | 21 | -11 | 1368.78 | 0.015 | 10618 |
| | Frontal (R) | Sub-gyral | 30 | 27 | 0 | 1321.91 | 0.017 | |
| | Frontal (R) | Inferior frontal gyrus | 32 | 36 | -11 | 1176.26 | 0.026 | |
| 4 | Sublobar (R) | Lentiform nucleus | 26 | 2 | -6 | 927.62 | 0.048 | 6.75 |
| 5 | Sublobar (R) | Lentiform nucleus | 27 | 0 | -5 | 926.90 | 0.048 | 6.75 |

Abbreviations: IAT, internet addiction tendency; L, left; MNI, Montreal Neurological Institute; R, right; TFCE, threshold-free cluster enhancement; WM, white matter.

significant areas are mostly included and overlapping with areas that had significant negative correlations between continuous scoring values for Young's IAT scale and rWMV. A corrected $p$ value threshold of $p < 0.05$ was used for all analyses.

The fMRI analysis revealed that there were no significant correlations. However, subjects who scored ≥50 using Young's IAT scale showed a tendency of greater brain activity during the 2-back task in a similar area of significant correlation between brain activity during the 2-back task and Young's IAT scale (mPFC, x, y, z = 3, 51, −6, 972 mm³ under the threshold of $p < 0.001$, uncorrected).

These results suggest similar but weaker tendencies of the correlations of the dichotomized value (Young's IAT scale ≥50 = 1, Young's IAT scale < 0) as compared with the significant correlations between the continuous score of Young's IAT scale and neuroimaging measures.

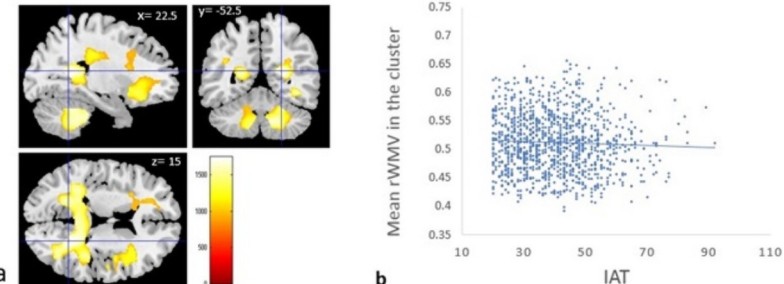

**Fig 4. Regional white matter volumes correlated with internet addiction tendency (IAT) score in young adults.** (a) The panels show the areas of significant negative correlation between IAT score and rWMV. The results shown were obtained using a threshold of threshold-free cluster enhancement (TFCE) of $p < 0.05$ based on 5,000 permutations. Significant correlations were found in the sub-gyral area of the temporal lobe, superior temporal gyrus, extra-nuclear, lentiform nucleus, right cerebellum anterior lobe, cerebellar tonsil, right inferior frontal gyrus, sub-gyral of frontal lobe, and pons. (b) Scatterplot of the association between IAT score and mean rWMV values of the largest cluster. The simple correlation coefficient between mean rWMV signal of the significant cluster and IAT score is −0.045. The association may look weak, but the partial correlation coefficient of this association when age, sex, and total intracranial volume were accounted for is − 0.108. IAT is negatively correlated with the mean rWMV of the significant cluster 1 (r = -0.113, $p$ = 0.003), significant cluster 2 (r = −0.108, $p$ = 0.005), and significant cluster 3 (r = -0.119, $p$ = 0.002) in men. In addition, IAT has a slight negative correlation with the mean rWMV in cluster 1 (-0.104, $p$ = 0.021) in women.

**Table 5. Brain regions exhibiting significant positive correlations with IAT score.**

| Anatomical area | MNI coordinates | | | TFCE value | Corrected p value (FWE) | Cluster size (mm³) | Activated areas, deactivated areas during the 2-back task* |
|---|---|---|---|---|---|---|---|
| | X | Y | Z | | | | |
| Left medial frontal gyrus | -9 | 54 | -3 | 737.44 | 0.014 | 23112 | 0%, 97.5% |
| Left superior frontal gyrus, medial part | -9 | 54 | 6 | 728.89 | 0.015 | | |
| Right anterior cingulate | 6 | 39 | 6 | 675.55 | 0.019 | | |

Abbreviations: IAT, internet addiction tendency; L, left; MNI, Montreal Neurological Institute; R, right; TFCE, threshold-free cluster enhancement; WM, white matter.
*Percentage of voxels showing significant activation or deactivation ($p < 0.05$, false discovery rate (FDR) corrected at the voxel level) during the 2-back task among the 63 subjects sampled, from which the template of the diffusion image was created [86].

**Supplemental ROI analyses of the associations between activity in key nodes of the DMN and IAT scores.** After correcting for multiple comparisons, the partial correlation analyses showed that the IAT score significantly and positively correlated with brain activity (2-back) of the ROI of the mPFC, left hippocampus, and right hippocampus. Similar tendencies were observed for the brain activity (2-back) of ROI of the PCC/precuneus, and for brain activity (0-back) in the ROI of the left and right hippocampus (S1 Table).

## Discussion

To the best of our knowledge, this is the first study to investigate the associations between internet addiction tendency and brain activity during a working memory task in healthy young adults. First, VBM showed a positive association of IAT score with GMV across the supramarginal gyrus and negative associations of IAT score with rWMVs in the right inferior frontal gyrus (rIFG) and sub-gyral frontal lobe, extra-nuclear, lentiform nucleus, right cerebellum anterior lobe, cerebellar tonsil, sub-gyral temporal lobe, superior temporal gyrus, and pons. These rWMV correlations with IAT score are consistent with our original hypothesis that IAT is strongly associated with abnormal brain structures in fronto-striatal areas [59, 60]. However, cortical areas outside the frontal lobe were significant.

In this study, the association between IAT scores and brain activity during the WM task was observed only in the anterior part of the DMN (the mPFC and contingent regions), but

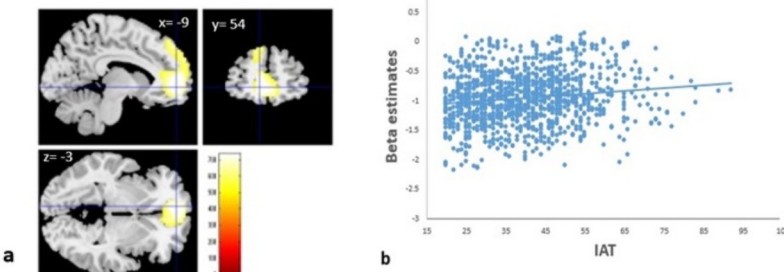

**Fig 5.** (a) Regional brain activity correlates with internet addiction tendency (IAT) scores. Regions with significant correlations between brain activity and IAT scores are overlaid on a single subject T1 image from SPM8. Results were obtained using a threshold of threshold-free cluster enhancement (TFCE) of $p < 0.05$ based on 5,000 permutations. IAT scores were significantly and positively correlated with brain activity during the 2-back task in the default-mode network (medial frontal gyrus and anterior cingulum). (b) Scatterplot of the relationship between the IAT scores and brain activity during the 2-back task in the default-mode network. IAT showing a positive correlation with regional brain activity in men (r = 0.113, $p = 0.003$) and in women (r = 0.177, $p = 0.001$).

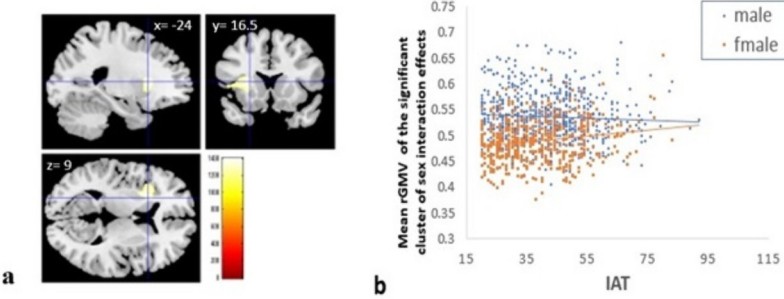

**Fig 6. Interaction between sex and internet addiction tendency (IAT) scores.** (a) Whole-brain analyses of rGMV show significant effects of interaction between sex and the score of Young's IAT scale. Results were obtained using a threshold of threshold-free cluster enhancement (TFCE) of $p < 0.05$ based on 5,000 permutations. The significant effects of interaction were found in the area around the left anterior insula and left lentiform. This interaction is positively correlated in women (r = 0.171, $p = 0.001$), and not correlated in men (r = −0.051, $p = 0.189$). (b) Scatterplot of the mean rGMV of the significant cluster of sex interaction effects in the left basal ganglia and left anterior insula.

not in the posterior DMN. We added a supplemental ROI analyses that investigated the brain activity of functional ROIs of important nodes based on the DMN and IAT scores (Supplemental Methods, Supplemental Results, and S1 Table). This analysis showed there was a significant positive correlation between brain activity defined by the IAT score and brain activity observed in the mPFC and bilateral hippocampus. In addition, brain activity of the PCC/precuneus during the 2-back task, also showed a marginally insignificant positive correlation. Thus, it is difficult to conclude that there is no correlation between the IAT score and brain activity of the posterior DMN. Whether this relatively weaker result for the posterior DMN is due to statistical fluctuation or other reasons remains unclear. However, previous studies have demonstrated that reduced TID in the DMN during working memory tasks in the elderly or psychiatric patients is seen in both the posterior and anterior parts of the DMN [44, 45]. Following our discussion, Moccia, Pettorruso [92] explained that the activity in some brain networks, including the ACC, is the basis of response inhibition in healthy individuals. Also, deficits in response inhibition in individuals with substance use disorders and gambling disorder and relapse have been shown in previous studies. This study improves our understanding of the common underlying neural mechanisms of IAT and other addictive behaviors within this conceptual framework.

Moreover, our supplemental analysis revealed the tendency of positive correlation between the mean brain activity of the cluster of significant correlation between the IAT score and brain activity in the mPFC in this study. We also showed the accuracy of the 2-back task after controlling for age, sex, and framewise displacement during the scan (partial correlation coefficient: −0.049, $p = 0.096$). However, reaction time during the 2-back task did not show such tendencies. These findings suggest that the TID in the anterior part of the DMN is also associated with cognitive processes during working memory.

In the present study, the GMV of rSMG was significantly positive correlated with IAT score, consistent with recent studies implicating the supramarginal gyrus in addiction [93, 94]. Also in accord with functions in addiction, this region is responsible for phonological processing [95] and our recent study revealed that frequent internet use in children is associated with a decrease in verbal intelligence [52]. In most previous studies, however, there was a negative correlation between GMV volume and addiction [35–39], while our current study found positive correlations between IAT and GMV in the left caudate. These discrepancies among studies

may be due to differences in sample groups. Our sample group included only university students, who are of above average intelligence and would use the internet more frequently for learning and education. Another possible explanation is greater internet accessibility in recent years (via smart phones, Wi-Fi, etc.). The possible mechanisms are diverse. For example, the development of the smartphone and accessibility Wi-Fi have made it easy to use the internet under the condition of dual tasks (e.g., engaging in the internet while walking), which might lead to changes in functions and structures of attention-related areas in the users. Faster internet speed might allow faster access to the verbal and visual information, which may in turn lead to structural changes in relevant brain areas in users. However, these explanations are speculative and require future study.

Our study also revealed negative correlations between the IAT score and rWMV in frontal, temporal and sublobar areas, regions responsible for response inhibition, visuospatial/visuomotor functions, and reward system functions. These findings are consistent with our hypothesis that IAT would correlate with WMVs in fronto-striatal areas. As stated, fronto-striatal circuits are critical for the emergence of addictive behaviors. Previous studies have demonstrated contributions of the right inferior frontal gyrus rIFG to addiction [96, 97] likely through critical functions in response inhibition, decision making, target detection, and inhibitory control [98]. Impulsive responses are inhibited by engaging frontal–basal ganglia pathways involving the rIFG, striatum, pre-supplementary motor area (pre-SMA), and subthalamic nucleus (STN) [99]. Previous studies have well documented the underlying role of the IFG in addiction [100].

This finding from our study suggests that the tendency to IA may have a common underpinning with other substance abuse disorders.

The cortex is connected to the subthalamic nucleus via a hyperdirect pathway as well as by a slower indirect pathway in which cortical outputs are first sent to the striatum, then passed to the globus pallidus pars externa, and finally to the STN [101]. Previous studies have shown that both the hyperdirect and indirect basal ganglia pathways are critical for response inhibition [99, 102]. Thus, negative correlations between IAT score and rWMVs of frontal, temporal, and sublobar areas may reflect poor response control for internet use.

There was also a negative correlation between IAT score and WMVs of right temporal lobe (sub-gyral and superior temporal gyrus). This result is consistent with previous findings indicating that addiction is associated with abnormalities in the cerebral cortex, including the temporal cortex. For instance, Fortier et al. [103] showed that alcoholism in adults is itself linked to a decrease in cortical thickness in the temporal, frontal, and occipital cortical regions and these changes correlated positively with the severity of abuse. Further, significant negative correlations between rWMVs and IAT scores were found in the sublobar regions and lobes of the cerebellum. Moulton, Elman [104] posited that the cerebellum, as an intermediary between motor function and reward, motivation, and cognitive control systems would have important roles in the etiology of addiction. Also, some studies showed a correlation between subjective craving among heroin dependents and brain activities in the superior temporal gyrus region [105].

These negative correlations between regional WMVs and IAT scores may reflect reduced myelination or loss of WM integrity within these pathways. As detailed in our previous studies [85, 86], changes in myelination, glial cell number, glial cell size, and the number of axon collaterals can all influence WMV. Therefore, decreased regional WMV may reflect reduced myelination, glial cell number/size, and (or) axonal number, which in turn impedes both regional neural transmission and neural transmission among networks. Thus, decreases in these physiological components in fronto-striatal pathways, right temporal lobe, sublobar regions,

cerebellum lobe, and ensuing transmission deficits may lead to impaired response inhibition and visuospatial/visuomotor and reward system dysfunction. The present findings thus advance our understanding of WM dysfunction in IAT among young adults. More intense IAT is associated with WMV reductions in core brain regions responsible for response inhibition, visuospatial/visuomotor functions, and reward.

It has been reported that addictive behaviors usually onset in young adult age. This is explained by several reasons, including dramatic physical, cognitive, and psychosocial changes occurring at that time [106]. So, the results of this study are also important and innovative in that our knowledge about the neurological underpinnings of addictive behaviors in young people increased.

## Conclusions

In conclusion, we demonstrate significant association of IAT severity with both white and gray matter volumes, as well as with DMN activity during a working memory task. Internet addiction tendency is characterized by increased gray matter volume in the rSMG brain region. This region is responsible for phonological processing, decreased rWMVs in brain regions involved in inhibition, visuospatial/visuomotor functions, and the reward system. Moreover, IAT is correlated with reduced TID of the DMN. Collectively, our findings suggest that IAT may share neural mechanisms with other types of addiction.

## Limitations and further research

This study has several limitations. First, the cross-sectional design precludes establishment of causal relationships between IAT and changes in specific brain structures and activity patterns during a WM task. Second, as this study cohort consisted only of healthy young adults at a relatively high educational level, these findings may be extrapolated to the general population. Age, intellectual ability, education level, and general health can also strongly influence brain structures and increase sensitivity of the analyses [107].

## Supporting information

**S1 Table. Supplemental ROI analyses of associations between activity in key nodes of the DMN and IAT scores.** Abbreviations: FDR, false discovery rate; unc, uncorrected; L, left; R, right; mPFC, medial prefrontal cortex; PCC, posterior cingulate cortex.
(DOCX)

**S1 Abbreviations.**
(DOCX)

## Acknowledgments

We respectfully thank Yuki Yamada for operating the MRI scanner and Haruka Nouchi for acting as an examiner for psychological tests. We also thank study participants, the other examiners of psychological tests, and all of our colleagues at the Institute of Development, Aging and Cancer, Tohoku University, for their support.

## Author Contributions

**Conceptualization:** Hikaru Takeuchi, Yasuyuki Taki, Ryuta Kawashima.

**Data curation:** Hikaru Takeuchi, Yasuyuki Taki, Rui Nouchi, Ryoichi Yokoyama, Yuka Kotozaki, Seishu Nakagawa, Atsushi Sekiguchi, Kunio Iizuka, Sugiko Hanawa, Tsuyoshi Araki,

Carlos Makoto Miyauchi, Kohei Sakaki, Takayuki Nozawa, Shigeyuki Ikeda, Susumu Yokota, Daniele Magistro, Yuko Sassa.

**Formal analysis:** Saeid Sadeghi, Hikaru Takeuchi, Yasuyuki Taki.

**Funding acquisition:** Hikaru Takeuchi, Yasuyuki Taki, Ryuta Kawashima.

**Investigation:** Hikaru Takeuchi, Yasuyuki Taki.

**Methodology:** Saeid Sadeghi, Hikaru Takeuchi, Yasuyuki Taki.

**Project administration:** Hikaru Takeuchi, Yasuyuki Taki, Ryuta Kawashima.

**Resources:** Hikaru Takeuchi, Yasuyuki Taki.

**Software:** Hikaru Takeuchi.

**Supervision:** Hikaru Takeuchi, Yasuyuki Taki.

**Validation:** Saeid Sadeghi, Hikaru Takeuchi, Bita Shalani.

**Visualization:** Saeid Sadeghi, Hikaru Takeuchi, Bita Shalani.

**Writing – original draft:** Saeid Sadeghi, Hikaru Takeuchi, Bita Shalani.

**Writing – review & editing:** Saeid Sadeghi, Hikaru Takeuchi, Bita Shalani.

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
