## [Decision Letter · Decision Letter 0]

16 Jul 2021

PONE-D-21-09795

Brain Structures and Activity During a Working Memory Task Associated with Internet Addiction Tendency in Young Adults: A Large Sample Study

PLOS ONE

Dear Dr. Sadeghi,

Thank you for submitting your manuscript to PLOS ONE. After careful consideration, we feel that it has merit but does not fully meet PLOS ONE’s publication criteria as it currently stands. Therefore, we invite you to submit a revised version of the manuscript that addresses the points raised during the review process.

The reviewers agreed that the manuscript addresses a timely and important topic. However, they suggested that the manuscript will benefit from  the improvement of English style and grammar, a more detailed description of socio-demographics characteristics (including the proportion  of participants that came from non-Japanese cultures), and a discussion of shared biological mechanisms between IAT and other types of addiction. The discussion section, in general, can be further improved by focusing more on whether and how  high IAT scores and associated structural and functional changes may alter functioning in affected individuals, rather then on the results reinstatement.   A more detailed description of the "laboratory’s routine questionnaire"  is necessary. Please explain why the IAT scores were split based at the score 50. Given that the lowest IAT score is 20, splitting the scale at 50 can bias the dichotomous distribution - the 'top' will have the range of 50 points, while the 'bottom' part will have the range of only 30 points. 

The 2-back RT of over 6 seconds (Table 1) seems unusually slow (approximately 10 times slower than that in the majority of other 2-back experiments). Please explain why that was the case and confirm that RT reported in Table 1 is in msec.

Please provide  image acquisition parameters  for all neuroimaging modalities including fMRI (nback) as well as all information related to the data preprocessing and analyses. All information related to this paper must be in the paper. There is not reason or need for redirecting readers to your previous publications. While using right-handed participants is customary in fMRI research, you may consider adding a very short justification for this recruitment strategy. Please explain what "using an appropriate threshold that successfully segregated each area in the representative DMN nodes" means and report the threshold that was deemed "appropriate". 

Please make sure that all abbreviations are explained in the text.

We look forward to receiving your revised manuscript.

Kind regards,

Anna Manelis, Ph.D.

Academic Editor

PLOS ONE

Journal Requirements:

2. Please change "female” or "male" to "woman” or "man" as appropriate, when used as a noun (see for instance https://apastyle.apa.org/style-grammar-guidelines/bias-free-language/gender).

"The authors declare no competing financial or non-financial interests."

We note that one or more of the authors are employed by a commercial company: ADVANTAGE Risk Management Co., Ltd.

4.1. Please provide an amended Funding Statement declaring this commercial affiliation, as well as a statement regarding the Role of Funders in your study. If the funding organization did not play a role in the study design, data collection and analysis, decision to publish, or preparation of the manuscript and only provided financial support in the form of authors' salaries and/or research materials, please review your statements relating to the author contributions, and ensure you have specifically and accurately indicated the role(s) that these authors had in your study. You can update author roles in the Author Contributions section of the online submission form.

4.2. Please also provide an updated Competing Interests Statement declaring this commercial affiliation along with any other relevant declarations relating to employment, consultancy, patents, products in development, or marketed products, etc.  

6. Please ensure that you include a title page within your main document. You should list all authors and all affiliations as per our author instructions and clearly indicate the corresponding author.

Reviewers' comments:

Reviewer's Responses to Questions

**Comments to the Author**

1. Is the manuscript technically sound, and do the data support the conclusions?

Reviewer #1: Yes

Reviewer #2: Yes

2. Has the statistical analysis been performed appropriately and rigorously? 

Reviewer #1: Yes

Reviewer #2: Yes

3. Have the authors made all data underlying the findings in their manuscript fully available?

Reviewer #1: No

Reviewer #2: Yes

4. Is the manuscript presented in an intelligible fashion and written in standard English?

Reviewer #1: No

Reviewer #2: Yes

5. Review Comments to the Author

Reviewer #1: Dear Authors,

topics covered in your article are of current interest since the field of Internet Addiction Tendency (IAT) is to date less studied as compared to Internet Addiction Disorder (IAD).

The overall quality of the article is good and the items are fluently addressed. However, a linguistic revision of style and grammar by a native-speaker would be useful (e.g. on page 13 the expression "who are of above average intelligence" should be made better as "high cultural level").

As for the section 'Materials and Methods', I would suggest specifying the temporal frame considered for "recent" use of psychoactive drugs, list inclusion and exclusion criteria in a more orderly fashion and eventually add a supplemental table with socio-demographics characteristics of the sample.

With the aim of improving the background of the study, specifically when referring to the association among working memory, impulsivity and ADHD, authors might benefit from having a look at "Di Nicola M, et al. Adult attention-deficit/hyperactivity disorder in major depressed and bipolar subjects: role of personality traits and clinical implications. Eur Arch Psychiatry Clin Neurosci. 2014 Aug;264(5):391-400. doi: 10.1007/s00406-013-0456-6." Further, in order to enrich the Discussion, I would suggest commenting on the following articles: “Moccia L, et al. Neural correlates of cognitive control in gambling disorder: a systematic review of fMRI studies. Neurosci Biobehav Rev. 2017 Jul;78:104-116. doi: 10.1016/j.neubiorev.2017.04.0252; Di Nicola M, et al. Gender Differences and Psychopathological Features Associated With Addictive Behaviors in Adolescents. Front Psychiatry. 2017 Dec 1;8:256. doi: 10.3389/fpsyt.2017.00256.” In this view, the study could be improved by deepening the underlying mechanisms potentially shared by IAT and other addictive behaviors, especially in the youngest.

Best regards.

Reviewer #2: I thank the authors and the section editor for bringing this work to my attention.

It is an interesting proposal “Brain Structures and Activity During a Working Memory Task Associated with Internet Addiction Tendency in Young Adults: A Large Sample Study”. but some clarification is needed:

1. Page 1, 2, 3, 4, 6, 7, 11 and 13; WM, fMRI, GABA, EPI, UK, TFCE, ACC and PCC were not clear. Write them in their long form for their first appearance. Please try to include them in the abbreviation session.

2. Page 4: Why do you prefer to use the right handed participants?

3. Page 5: The IAT scale minimum and maximum scores are 20 and 100, with higher scores reflecting a greater tendency toward internet addiction. The Japanese version of this scale has demonstrated high reliability and validity. What does a higher score of internet addiction test mean? It should be specified clearly based on their degree of severity. As you stated your study participants were students from Tohoku University in Japan and you might have international students out of Japan with different culture, religion and so on. So, have you done a validity test for your study?

4. Page 8: The interaction between sex and the score of Young’s IAT scale (contrasts of [the score of the Young IAT scale for males, the effect of the score of the Young IAT scale for females] were [−1 1] or [1 –1]) were assessed using t-contrasts. Correction for multiple comparisons was performed using the same method used in the whole-brain multiple regression analyses. There is repetition of words as highlighted above. So, remove one of them.

5. Page 9 and Page 11: For this comparison, we conducted multiple regression analyses in which all dependent and independent variables of the main analyses remained the same, except that Young’s IAT score was replaced by the dichotomized value (Young’s IAT scale ≥50 = 1, Young’s IAT scale < 0).What does it mean?

6. PLOS authors have the option to publish the peer review history of their article (what does this mean?). If published, this will include your full peer review and any attached files.

Reviewer #1: No

Reviewer #2: No

---

## [Author Response · Author response to Decision Letter 0]

14 Sep 2021

Response to Reviewers

Manuscript ID: PONE-D-21-09795

Manuscript title: Brain Structures and Activity During a Working Memory Task Associated with Internet Addiction Tendency in Young Adults: A Large Sample Study

Journal: PLOS ONE

Dear editor and reviewers,

We sincerely thank the editor and reviewers for constructive and valuable comments, which were of great help in revising the manuscript. Accordingly, the revised manuscript has been systematically improved with new information and additional interpretations. Our responses to the editor comments (ARo), first referee’s comments (AR1), second referee’s comments (AR2), and journal requirements (ARjr) are given below. Also, green text indicating changes has been added to the revised manuscript.

Editor comments

EC 1. The manuscript will benefit from the improvement of English style and grammar.

AR0 1: Thanks for your suggestion. The manuscript reviewed by Enago (www.enago.jp) from the improvement of English style and grammar. 

EC 2. The manuscript will benefit from a more detailed description of socio-demographics characteristics (including the proportion of participants that came from non-Japanese cultures)

AR0 2: Thank you for your valuable suggestion. This point added to the revised manuscript. Also, according to your and Reviewer #1 suggestion, more detailed description of socio-demographics characteristics of the sample presented in a new table. as follows:

‘All subjects were students from Tohoku University and neighboring universities and colleges in Japan. All but one of the subjects in this study were native Japanese speakers. However, one foreign Asian student who was very proficient in Japanese and was determined to be equipped to go through the experimental Japanese procedures like native Japanese speakers was allowed to participate in this study. The removal of this one subject affects the results little.. … .

The participant’s socio-demographic characteristics are presented in Table 1.

Table 1: The socio-demographic characteristics of participants

Variable Min Max M SD

Age (year) 18 27 20.71 1.78

Self-reported height 142 192 166.35 8.44

Self-reported weight 38 115 57.96 9.50

BMI 15.39 32.88 20.83 1.74

Family annual income a 1 7 4.19 1.56

parent years of education b 9 21 14.69 1.85

a Family annual income was classified as follows: 1, annual income below 2 million yen; 2, 2–4 million yen; 3, 4–6 million yen; 4, 6–8 million yen; 5, 8–10 million yen; 6, 10–12 million yen; 7, >12 million yen; the currency exchange rate is approximately $1 USD = 120 yen.

b Parent average educational qualification (years of education) was classified as follows: 6 years, elementary school graduate or below; 9 years, junior high school graduate; 11 years, graduate of a short-term school completed after junior high school; 12 years, normal high school graduate; 14 years, graduate of a short-term school completed after high school (such as a junior college); 16 years, university graduate; 18 years, Master’s degree; and 21 years, doctorate.

EC 3. The manuscript will benefit from a discussion of shared biological mechanisms between IAT and other types of addiction. The discussion section, in general, can be further improved by focusing more on whether and how high IAT scores and associated structural and functional changes may alter functioning in affected individuals, rather than on the results reinstatement. 

AR0 3: We appreciate your constructive suggestion. We tried to improve the discussion of the article according to your comments. The corrections and additions made in the discussion are highlighted in green.

Following our discussion, Moccia, Pettorruso [97] explained that the activity in some brain networks, including the ACC, is the basis of response inhibition in healthy individuals. Also, deficits in response inhibition in individuals with substance use disorders (SUD) and gambling disorder (GD) and relapse have been shown in previous studies. Thus, this study improves our understanding of the common underlying neural mechanisms of IAT and other addictive behaviors within this conceptual framework. 

Previous studies have well documented the underlying role of the IFG in addiction [104].

This finding from our study suggests that the tendency to internet addiction may have a common underpinning with other substance abuse disorders.

Also, some studies showed a correlation between subjective craving among heroin dependents and brain activities in the superior temporal gyrus region [109].

These negative correlations between regional WMVs and IAT scores may reflect reduced myelination or loss of WM integrity within these pathways. As detailed in our previous studies [91, 110], changes in myelination, glial cell number, glial cell size, and the number of axon collaterals can all influence WMV. Therefore, decreased regional WMV may reflect reduced myelination, glial cell number/size, and (or) axonal number, which in turn impedes both regional neural transmission and neural transmission among networks. Thus, decreases in these physiological components in fronto-striatal pathways, right temporal lobe, sublobar regions, cerebellum lobe, and ensuing transmission deficits may lead to impaired response inhibition and visuospatial/visuomotor and reward system dysfunction. The present findings thus advance our understanding of WM dysfunction in IAT among young adults. More intense IAT is associated with WMV reductions in core brain regions responsible for response inhibition, visuospatial/visuomotor functions, and reward.

It has been reported that addictive behaviors usually onset in young adult age. This is explained by several reasons, including dramatic physical, cognitive, and psychosocial changes occurring at that time [111]. So, the results of this study are also significant and innovative in that our knowledge about the neurological underpinnings of addictive behaviors in young people increased.

EC 4. A more detailed description of the "laboratory’s routine questionnaire” is necessary. 

AR0 4: Thanks for your attention. We added more detailed description of the "laboratory’s routine questionnaire” as follow:

A history of psychiatric and neurological diseases and/or recent drug use was assessed using our laboratory’s routine questionnaire, in which each subject answered questions related to their current or previous experiences of any of the listed diseases and listed drugs that they had recently taken. The questionnaire also asked the personal contact information, age, birthday, the institutes they belong to, age, sex, weight, and height. The Edinburgh Handedness Inventory [1] was also included in this routine questionnaire.

EC 5. Please explain why the IAT scores were split based at the score 50. Given that the lowest IAT score is 20, splitting the scale at 50 can bias the dichotomous distribution - the 'top' will have the range of 50 points, while the 'bottom' part will have the range of only 30 points. 

AR0 5: We would like to thank you for your attention. According to recently conducted studies, a score of ≥ 50 in Young' IAT scale was considered as problematic internet use [2, 3]. This study included a number of participants who scored ≥ 50 in Young' IAT scale. Also, previous reviewer (in another journal) suggested that we do this analysis. The referee's argument was that a comprehensive analysis of continuous and dichotomous may produce more convincing findings. So, we have added a supplement analysis to the main analysis.

EC 6. The 2-back RT of over 6 seconds (Table 1) seems unusually slow (approximately 10 times slower than that in the majority of other 2-back experiments). Please explain why that was the case and confirm that RT reported in Table 1 is in msec.

AR0 6: Thank you for your attention. 6729 means 0.6729sec. We are so sorry for confusing values. The values are revised in Table.

EC 7. Please provide image acquisition parameters for all neuroimaging modalities including fMRI (nback) as well as all information related to the data preprocessing and analyses. 

AR0 7: Thank you for your suggestion. We added parameters as follow:

The MRI acquisition methods were described in our previous study [4]. Briefly, all MRI data were acquired using a 3 Tesla (3T) Philips Achieva scanner. Diffusion-weighted data were acquired using a spin-echo Echo planar imaging (EPI) sequence [repetition time (TR) = 10293 millisecond (ms), echo time (TE) = 55 ms, field-of-view (FOV) = 22.4 centimeter (cm), 2×2×2 millimeter (mm)3 voxels, 60 slices, sensitivity encoding (SENSE) reduction factor = 2, number of acquisitions = 1]. The diffusion weighting was isotropically distributed along 32 directions (b value = 1,000 s/mm2). In addition, three images with no diffusion weighting (b value = 0 s/mm2 or b = 0 images) were acquired using a spin-echo EPI sequence (TR = 10293 ms, TE = 55 ms, FOV = 22.4 cm, 2 � 2 � 2 mm3 voxels, 60 slices). For the n-back session, 174 functional volumes were obtained [5]. High-resolution T1-weighted structural images were collected using a magnetization-prepared rapid gradient echo sequence (T1WIs: 240 × 240 matrix, TR = 6.5 ms, TE = 3 ms, FOV = 24 cm, slices = 162, slice thickness = 1.0 mm).

…

The following procedures for functional activation data analysis were reproduced from our previous study, as described previously [88]. From the images collected, fractional anisotropy (FA) and mean diffusion (MD) maps were calculated. [89]. In current study, these FA and MD maps were used during preprocessing of BOLD images. Prior to analysis, individual BOLD images were re-aligned and resliced to the mean BOLD image and 

then corrected for slice timing. Also, the abovementioned mean BOLD image was then realigned to the mean b = 0 image as previously described together with slice timing corrected images [80]. As the mean b = 0 image was aligned with the FA image and MD map, the BOLD image, b = 0 image, FA image, and MD map were all aligned.

All images were subsequently normalized using a previously validated two-step “new segmentation” algorithm of diffusion images and a previously validated diffeomorphic anatomical registration through exponentiated lie algebra (DARTEL)-based registration process [90]. This normalization method was used for all diffusion images, including gray matter segments (regional gray matter density [rGMD] map), white matter segments (regional white matter density [rWMD] map), and cerebrospinal fluid (CSF) segments (regional CSF density [rCSFD] map). Using the newly implemented segmentation algorithm in SPM8, the FA images of each individual were segmented into six tissues (first new segment). In this process, default parameters and tissue probability maps were used, except that affine regularization was performed using the International Consortium for Brain Mapping (ICBM) template for East Asian brains, and the sampling distance (the approximate distance between sampled points when estimating the model parameters) was 2 mm.

Next, we synthesize the FA image and the MD map. In this synthesized image, the area with WM tissue probability > 0.5 in the aforementioned new segmentation process was the FA image multiplied by −1. Hence, this synthesized image shows a very clear contrast between WM and other tissues. The remaining area is the MD map. 

We continued with the DARTEL registration process implemented in SPM8. During this process, we used the DARTEL import image of the GM tissue probability map produced by the second new segmentation process as the GM input for the DARTEL process. First, the raw FA image was multiplied by the WM tissue probability map from the second new segmentation process within the areas with a WM probability > 0.5 (the signals from all other areas were set to 0). Then, this FA image×the WM tissue probability map was coregistered and resliced to the DARTEL import WM tissue probability image from the second segmentation. The template for the DARTEL procedures was generated using imaging data from the 63 subjects who participated in [80] and in the present study. Then, using this existing template, DARTEL procedures were conducted. The parameters have been changed as follows to improve the accuracy of the procedures. The number of Gauss–Newton iterations to be performed within each outer iteration was set to 10. In each outer iteration, we used 8-fold more timepoints than the default values to solve the partial differential equations. The number of cycles used by the full multi-grid matrix solver was set to 8. The number of relaxation iterations performed in each multi-grid cycle was also set to 8. The resultant synthesized images were spatially normalized to Montreal Neurological Institute (MNI) space. The voxel size of the normalized BOLD image is 3 3 3 mm3.

EC 8. All information related to this paper must be in the paper. There is not reason or need for redirecting readers to your previous publications. 

AR0 8: We try to report all information related to this paper in the revised manuscript as follow:

The participant’s socio-demographic characteristics are presented in Table 1.

High-resolution T1-weighted structural images were collected using a magnetization-prepared rapid gradient echo sequence (T1WIs: 240 × 240 matrix, TR = 6.5 ms, TE = 3 ms, FOV = 24 cm, slices = 162, slice thickness = 1.0 mm). 

…

The following procedures for functional activation data analysis were reproduced from our previous study, as described previously [88]. From the images collected, fractional anisotropy (FA) and mean diffusion (MD) maps were calculated. [89]. In current study, these FA and MD maps were used during preprocessing of BOLD images. Prior to analysis, individual BOLD images were re-aligned and resliced to the mean BOLD image and 

then corrected for slice timing. Also, the abovementioned mean BOLD image was then realigned to the mean b = 0 image as previously described together with slice timing corrected images [80]. As the mean b = 0 image was aligned with the FA image and MD map, the BOLD image, b = 0 image, FA image, and MD map were all aligned.

All images were subsequently normalized using a previously validated two-step “new segmentation” algorithm of diffusion images and a previously validated diffeomorphic anatomical registration through exponentiated lie algebra (DARTEL)-based registration process [90]. This normalization method was used for all diffusion images, including gray matter segments (regional gray matter density [rGMD] map), white matter segments (regional white matter density [rWMD] map), and cerebrospinal fluid (CSF) segments (regional CSF density [rCSFD] map). Using the newly implemented segmentation algorithm in SPM8, the FA images of each individual were segmented into six tissues (first new segment). In this process, default parameters and tissue probability maps were used, except that affine regularization was performed using the International Consortium for Brain Mapping (ICBM) template for East Asian brains, and the sampling distance (the approximate distance between sampled points when estimating the model parameters) was 2 mm.

Next, we synthesize the FA image and the MD map. In this synthesized image, the area with WM tissue probability > 0.5 in the aforementioned new segmentation process was the FA image multiplied by −1. Hence, this synthesized image shows a very clear contrast between WM and other tissues. The remaining area is the MD map. 

We continued with the DARTEL registration process implemented in SPM8. During this process, we used the DARTEL import image of the GM tissue probability map produced by the second new segmentation process as the GM input for the DARTEL process. First, the raw FA image was multiplied by the WM tissue probability map from the second new segmentation process within the areas with a WM probability > 0.5 (the signals from all other areas were set to 0). Then, this FA image×the WM tissue probability map was coregistered and resliced to the DARTEL import WM tissue probability image from the second segmentation. The template for the DARTEL procedures was generated using imaging data from the 63 subjects who participated in [80] and in the present study. Then, using this existing template, DARTEL procedures were conducted. The parameters have been changed as follows to improve the accuracy of the procedures. The number of Gauss–Newton iterations to be performed within each outer iteration was set to 10. In each outer iteration, we used 8-fold more timepoints than the default values to solve the partial differential equations. The number of cycles used by the full multi-grid matrix solver was set to 8. The number of relaxation iterations performed in each multi-grid cycle was also set to 8. The resultant synthesized images were spatially normalized to Montreal Neurological Institute (MNI) space. The voxel size of the normalized BOLD image is 3 3 3 mm3.

EC 9. While using right-handed participants is customary in fMRI research, you may consider adding a very short justification for this recruitment strategy. 

AR0 9: Thank you for your suggestion. We added a short justification:

We used the Edinburgh Handedness Inventory to evaluate handedness in subjects. Previous studies demonstrated significant differences in brain morphology and activity patterns between right‐handers and left‐handers [6-10]. For this reason, fMRI studies tend to exclude left-handers.

EC 10. Please explain what "using an appropriate threshold that successfully segregated each area in the representative DMN nodes" means and report the threshold that was deemed "appropriate". 

AR0 10: We meant: “T score threshold” is the appropriate threshold for each ROI.

ROIs were medial prefrontal cortex (mPFC) (peak coordinate: x = −6, y = 57, z = −6, T score threshold = 15, 425 voxels), posterior cingulate cortex (PCC)/precuneus (peak coordinate: x = −6, y = −57, z = 12, T score threshold = 15, 305 voxels), left hippocampus (peak coordinate: x = −27, y = −21, z = −24, T score threshold = 9, 27 voxels), right hippocampus (peak coordinate: x = 24, y = −15, z = −27, T score threshold = 9, 66 voxels), left temporoparietal junction (peak coordinate: x = −45, y = −72, z = 21, T score threshold = 7, 322 voxels), and right temporoparietal junction (peak coordinate: x = 54, y = −69, z = 27, T score threshold = 7, 32 voxels).

EC 11. Please make sure that all abbreviations are explained in the text.

 AR0 11: Thank you very much for your suggestion. We have reviewed and completed all the abbreviations in the text. Also, these abbreviations were included in a separate Word file (Abbreviation file) to be included in the appropriate selection of the manuscript at the discretion of the editor.

Review Comments to the Author 

Reviewer #1:

RC1 1: The overall quality of the article is good and the items are fluently addressed. However, a linguistic revision of style and grammar by a native-speaker would be useful (e.g. on page 13 the expression "who are of above average intelligence" should be made better as "high cultural level").

AR1 1: The manuscript reviewed by Enago (www.enago.jp).

RC1 2: As for the section 'Materials and Methods', I would suggest specifying the temporal frame considered for "recent" use of psychoactive drugs.

 AR1 2: Thank you for your attention and sorry for the inappropriate wording. The correct term is the “current use of the psychoactive drugs”. We revised this in the manuscript.

RC1 3: list inclusion and exclusion criteria in a more orderly fashion and eventually add a supplemental table with socio-demographics characteristics of the sample.

AR1 3: Many thanks for your suggestion. We added the details of recruitment and exclusion criteria of subjects as bellow and add a table (Table 1) with socio-demographics characteristics of the sample.

All subjects were students from Tohoku University and neighboring universities and colleges in Japan. All but one of the subjects in this study were native Japanese speakers. However, one foreign Asian student who was very proficient in Japanese and was determined to be equipped to go through the experimental Japanese procedures like native Japanese speakers was allowed to participate in this study. The removal of this one subject affects the results little.They were recruited using advertisements on bulletin boards at Tohoku University or via e-mail introducing the study. These advertisements and e-mails specified the exclusionary characteristics for individuals regarding participation in the study, such as handedness, the existence of metal in and around the body, claustrophobia, the use of certain drugs, and a history of certain psychiatric disorders and neurological diseases, and previous participation in related experiments. A history of psychiatric and neurological diseases and/or recent drug use was assessed using our laboratory’s routine questionnaire, in which each subject answered questions related to their current or previous experiences of any of the listed diseases and listed drugs that they had recently taken. The questionnaire also asked the personal contact information, age, birthday, the institutes they belong to, age, sex, weight, and height. Edinburgh Handedness Inventory [1] is also included in this routine questionnaire. We used the Edinburgh Handedness Inventory to evaluate handedness in subjects. Previous studies demonstrated that there are significant differences in brain morphology and activity patterns between right‐handers and left‐handers [74-78]. For this reason, fMRI studies tend to exclude left-handers. In the course of this experiment, the scans were checked for noticeable brain lesions and tumors, but no subjects had such obvious lesions or tumors. These descriptions were mainly obtained from our previously published work [11]. The participant’s socio-demographic characteristics are presented in Table 1. 

Table 1: The socio-demographic characteristics of participants

Variable Min Max M SD

Age (year) 18 27 20.71 1.78

Self-reported height 142 192 166.35 8.44

Self-reported weight 38 115 57.96 9.50

BMI 15.39 32.88 20.83 1.74

Family annual income a 1 7 4.19 1.56

parent years of education b 9 21 14.69 1.85

a Family annual income was classified as follows: 1, annual income below 2 million yen; 2, 2–4 million yen; 3, 4–6 million yen; 4, 6–8 million yen; 5, 8–10 million yen; 6, 10–12 million yen; 7, >12 million yen; the currency exchange rate is approximately $1 USD = 120 yen.

b Parent average educational qualification (years of education) was classified as follows: 6 years, elementary school graduate or below; 9 years, junior high school graduate; 11 years, graduate of a short-term school completed after junior high school; 12 years, normal high school graduate; 14 years, graduate of a short-term school completed after high school (such as a junior college); 16 years, university graduate; 18 years, Master’s degree; and 21 years, doctorate.

RC1 4: With the aim of improving the background of the study, specifically when referring to the association among working memory, impulsivity and ADHD, authors might benefit from having a look at "Di Nicola M, et al. Adult attention-deficit/hyperactivity disorder in major depressed and bipolar subjects: role of personality traits and clinical implications. Eur Arch Psychiatry Clin Neurosci. 2014 Aug;264(5):391-400. doi: 10.1007/s00406-013-0456-6." 

Further, in order to enrich the Discussion, I would suggest commenting on the following articles: “Moccia L, et al. Neural correlates of cognitive control in gambling disorder: a systematic review of fMRI studies. Neurosci Biobehav Rev. 2017 Jul;78:104-116. doi: 10.1016/j.neubiorev.2017.04.0252; 

Di Nicola M, et al. Gender Differences and Psychopathological Features Associated With Addictive Behaviors in Adolescents. Front Psychiatry. 2017 Dec 1;8:256. doi: 10.3389/fpsyt.2017.00256.”

In this view, the study could be improved by deepening the underlying mechanisms potentially shared by IAT and other addictive behaviors, especially in the youngest. 

AR1 4: Many thanks for your valuable suggestion. We considered these interesting studies. We referenced Di Nicola M, et al. study in the text.

Following our discussion, Moccia, Pettorruso [12] explained that the activity in some brain networks, including the ACC, is the basis of response inhibition in healthy individuals. Also, deficits in response inhibition in individuals with substance use disorders (SUD) and gambling disorder (GD) and relapse have been shown in previous studies. This study improves our understanding of the common underlying neural mechanisms of IAT and other addictive behaviors within this conceptual framework. 

It has been reported that addictive behaviors usually onset in young adult age. This is explained by several reasons, including dramatic physical, cognitive, and psychosocial changes occurring at that time [13]. So, the results of this study are also significant and innovative in that our knowledge about the neurological underpinnings of addictive behaviors in young people increased.

Reviewer #2

RC2 1. Page 1, 2, 3, 4, 6, 7, 11 and 13; WM, fMRI, GABA, EPI, UK, TFCE, ACC and PCC were not clear. Write them in their long form for their first appearance. Please try to include them in the abbreviation session.

AR2 1: Thank you very much for your attention and suggestion. We have reviewed and completed all the abbreviations in the text. Also, these abbreviations were included in a separate Word file (Abbreviation file) to be included in the selection of the article at the discretion of the magazine editor.

RC2 2. Page 4: Why do you prefer to use the right-handed participants?

AR2 2: Thank you for your question. We added a justification for this recruitment strategy:

Previous studies demonstrated significant differences in brain morphology and activity patterns between right‐handers and left‐handers [6-9]. For this reason, using right-handed participants is customary in fMRI research.

RC2 3. Page 5: The IAT scale minimum and maximum scores are 20 and 100, with higher scores reflecting a greater tendency toward internet addiction. The Japanese version of this scale has demonstrated high reliability and validity. What does a higher score of internet addiction test mean? It should be specified clearly based on their degree of severity. 

AR2 3: Many thanks for your comment. We rewrite this section as bellow:

Internet Addiction Tendency assessment

We used the Japanese version of Young’s IAT scale to assess condition severity [14]. This IAT instrument consists of 20 items answered on a 1–5 scale from 1 = “rarely” to 5 = “always”. The scale is self-administered and requires 5 to 10 minutes IAT measures the impact of internet use on people's daily routine, social life, productivity, sleeping pattern, and feelings. The IAT scale minimum and maximum scores are 20 and 100, with higher scores reflecting a greater tendency toward internet addiction. The developer of this scale suggests that a score of 20–39 points is an average online user who has complete control over his/her usage; a score of 40–69 signifies frequent problems due to internet usage, and a score of 70–100 means that the internet is causing significant problems [14]. The Japanese version of this scale has demonstrated high reliability and validity [15].

RC2 4. As you stated your study participants were students from Tohoku University in Japan and you might have international students out of Japan with different culture, religion and so on. So, have you done a validity test for your study?

AR2 4: We appreciate your attention. Since our assessment tools, instructions, and questionnaires were all in Japanese. All but one of the subjects in this study were native Japanese speakers. However, one foreign Asian student who was very proficient in Japanese and was determined to be equipped to go through the experimental Japanese procedures like native Japanese speakers was allowed to participate in this study. We added this point in the text of the manuscript. Please see below:

All subjects were students from Tohoku University and neighboring universities and colleges in Japan. All but one of the subjects in this study were native Japanese speakers. However, one foreign Asian student who was very proficient in Japanese and was determined to be equipped to go through the experimental Japanese procedures like native Japanese speakers was allowed to participate in this study. The removal of this one subject affects the results little.

RC2 5. Page 8: The interaction between sex and the score of Young’s IAT scale (contrasts of [the score of the Young IAT scale for males, the effect of the score of the Young IAT scale for females] were [−1 1] or [1 –1]) were assessed using t-contrasts. Correction for multiple comparisons was performed using the same method used in the whole-brain multiple regression analyses. There is repetition of words as highlighted above. So, remove one of them.

AR2 5: Thanks again for the reviewer’s attention and we apologize for this unintentional mistake. We corrected this. As follows:

The interaction between sex and scores on Young’s IAT scale was assessed using t-contrasts. Correction for multiple comparisons was performed using the same method used in the whole-brain multiple regression analyses.

RC2 6. Page 9 and Page 11: For this comparison, we conducted multiple regression analyses in which all dependent and independent variables of the main analyses remained the same, except that Young’s IAT score was replaced by the dichotomized value (Young’s IAT scale ≥50 = 1, Young’s IAT scale <50 = 0). What does it mean?

AR2 6. According to recently conducted studies, a score of ≥ 50 in Young' IAT scale was considered as problematic internet use [2, 3]. This study included a number of participants who scored ≥ 50 in Young' IAT scale. A comprehensive analysis of continuous and dichotomous usually produce more convincing findings. So, we have added a supplement analysis to the main analysis.

The above phrase that you asked about, in fact is the explanation for dichotomized analyses we have conducted. 

Journal Requirements: 

JR 1. Please ensure that your manuscript meets PLOS ONE's style requirements, including those for file naming. The PLOS ONE style templates can be found at

 ARjr 1: We appreciate your guidance. Done.

JR 2. Please change "female” or "male" to "woman” or "man" as appropriate, when used as a noun (see for instance https://apastyle.apa.org/style-grammar-guidelines/bias-free-language/gender).

 ARjr 2: Thank you for your suggestion. We revised this points.

JR 3. We note that the grant information you provided in the ‘Funding Information’ and ‘Financial Disclosure’ sections do not match.

 ARjr 3: Thank you so much for your attention. We revised.

We added funding information in title page file, as follow:

This study was supported by a Grant-in-Aid for Young Scientists (B) (KAKENHI 23700306) and a Grant-in-Aid for Young Scientists (A) (KAKENHI 25700012) from the Ministry of Education, Culture, Sports, Science, and Technology.

The funder provided support in the form of salaries for authors H.T., but did not have any additional role in the study design, data collection, and analysis, decision to publish, or preparation of the manuscript. The specific roles of these authors are articulated in the ”author contributions” section.

The authors declare no competing interests. 

JR 4. Thank you for stating the following in the Competing Interests section:

"The authors declare no competing financial or non-financial interests."

We note that one or more of the authors are employed by a commercial company: ADVANTAGE Risk Management Co., Ltd.

 ARjr 4: Tsuyoshi Araki belonged to the lab while we were doing the experiment, but he had left from the university and works in that company. No financial relationship exists between this study, and our lab. 

We have double checked authors’ role and have added the following passage to the section regarding funding in title page file:

Tsuyoshi Araki belonged to the ”Division of Developmental Cognitive Neuroscience, IDAC, Tohoku University” while we were doing the experiment, but he left the university and now works in the ADVANTAGE Risk Management Co., Ltd. This company did not have any additional role in the study design, data collection, analysis, decision to publish, or manuscript preparation.

JR 4.1. Please provide an amended Funding Statement declaring this commercial affiliation, as well as a statement regarding the Role of Funders in your study. If the funding organization did not play a role in the study design, data collection and analysis, decision to publish, or preparation of the manuscript and only provided financial support in the form of authors' salaries and/or research materials, please review your statements relating to the author contributions, and ensure you have specifically and accurately indicated the role(s) that these authors had in your study. You can update author roles in the Author Contributions section of the online submission form.

 ARjr 4.1: We revised funding information as follow:

This study was supported by a Grant-in-Aid for Young Scientists (B) (KAKENHI 23700306) and a Grant-in-Aid for Young Scientists (A) (KAKENHI 25700012) from the Ministry of Education, Culture, Sports, Science, and Technology.

The funder provided support in the form of salaries for authors H.T., but did not have any additional role in the study design, data collection, analysis, decision to publish, or manuscript preparation. The specific roles of these authors are articulated in the ”author contributions” section.

JR 4.2. Please also provide an updated Competing Interests Statement declaring this commercial affiliation along with any other relevant declarations relating to employment, consultancy, patents, products in development, or marketed products, etc. 

Please know it is PLOS ONE policy for corresponding authors to declare, on behalf of all authors, all potential competing interests for the purposes of transparency. PLOS defines a competing interest as anything that interferes with, or could reasonably be perceived as interfering with, the full and objective presentation, peer review, editorial decision-making, or publication of research or non-research articles submitted to one of the journals. Competing interests can be financial or non-financial, professional, or personal. Competing interests can arise in relationship to an organization or another person. Please follow this link to our website for more details on competing interests: http://journals.plos.org/plosone/s/competing-interests.

 ARjr 4.2: Thank you for your attention. WE include both an updated Funding Statement and Competing Interests Statement in our cover letter.

JR 5. We note that you have indicated that data from this study are available upon request. PLOS only allows data to be available upon request if there are legal or ethical restrictions on sharing data publicly. For information on unacceptable data access restrictions, please see http://journals.plos.org/plosone/s/data-availability#loc-unacceptable-data-access-restrictions.

 ARjr 5: All the experimental data obtained in the experiment of this study will be available to ones that were admitted in the ethics committee of Tohoku University, school of medicine. All the data sharing should be first admitted by the ethics committee of Tohoku University, school of medicine.

The contact information of the ethics committee is as follows (* should be replaced by @).

med-kenkyo*grp.tohoku.ac.jp

JR 6. Please ensure that you include a title page within your main document. You should list all authors and all affiliations as per our author instructions and clearly indicate the corresponding author.

 ARjr 6: Thank you for your comment. We reviewd authors list..

JR 7. Please include captions for your Supporting Information files at the end of your manuscript, and update any in-text citations to match accordingly. Please see our Supporting Information guidelines for more information: http://journals.plos.org/plosone/s/supporting-information.

ARjr 7: Many thanks for your information. We included captions for our Supporting Information files.

1. Oldfield, R.C., The assessment and analysis of handedness: the Edinburgh inventory. Neuropsychologia, 1971. 9(1): p. 97-113.

2. Gorgich, E.A.C., et al., Evaluation of internet addiction and mental health among medical sciences students in the southeast of Iran. Shiraz E Medical Journal, 2018. 19(1).

3. Santos, V., et al., Treatment outcomes in patients with Internet Addiction and anxiety. MedicalExpress, 2017. 4.

4. Takeuchi, H., et al., The association between resting functional connectivity and creativity. Cerebral Cortex, 2012. 22(12): p. 2921-2929.

5. Takeuchi, H., et al., Failing to deactivate: the association between brain activity during a working memory task and creativity. Neuroimage, 2011. 55(2): p. 681-687.

6. Jang, H., et al., Are there differences in brain morphology according to handedness? Brain and behavior, 2017. 7(7): p. e00730.

7. Cuzzocreo, J.L., et al., Effect of handedness on fMRI activation in the medial temporal lobe during an auditory verbal memory task. Human brain mapping, 2009. 30(4): p. 1271-1278.

8. Gao, Q., et al., Effect of handedness on brain activity patterns and effective connectivity network during the semantic task of Chinese characters. Scientific reports, 2015. 5(1): p. 1-11.

9. Jörgens, S., et al., Handedness and functional MRI-activation patterns in sentence processing. Neuroreport, 2007. 18(13): p. 1339-1343.

10. Bailey, L.M., L.E. McMillan, and A.J. Newman, A sinister subject: Quantifying handedness‐based recruitment biases in current neuroimaging research. European Journal of Neuroscience, 2020. 51(7): p. 1642-1656.

11. Takeuchi, H., et al., Sex-Dependent Effects of the APOE ɛ4 Allele on Behavioral Traits and White Matter Structures in Young Adults. Cerebral Cortex, 2021. 31(1): p. 672-680.

12. Moccia, L., et al., Neural correlates of cognitive control in gambling disorder: a systematic review of fMRI studies. Neuroscience & Biobehavioral Reviews, 2017. 78: p. 104-116.

13. Di Nicola, M., et al., Gender differences and psychopathological features associated with addictive behaviors in adolescents. Frontiers in psychiatry, 2017. 8: p. 256.

14. Young, K.S., Caught in the net: How to recognize the signs of internet addiction--and a winning strategy for recovery. 1998: John Wiley & Sons.

15. Osada, H., Internet addiction in Japanese college students: Is Japanese version of Internet Addiction Test (JIAT) useful as a screening tool. Bulletin of Senshu University School of Human Sciences, 2013. 3(1): p. 71-80.

---

## [Decision Letter · Decision Letter 1]

30 Sep 2021

PONE-D-21-09795R1Brain Structures and Activity During a Working Memory Task Associated with Internet Addiction Tendency in Young Adults: A Large Sample Study

PLOS ONE

Dear Dr. Sadeghi,

Thank you for submitting your manuscript to PLOS ONE. After careful consideration, we feel that it has merit but does not fully meet PLOS ONE’s publication criteria as it currently stands. Therefore, we invite you to submit a revised version of the manuscript that addresses the points raised during the review process. The reviewers are satisfied with the revised version of the manuscript. however, the manuscript needs minor changes that are mostly related to formatting.Per PLOS One regulations: **"****Do not include funding sources in the Acknowledgments or anywhere else in the manuscript file. Funding information should only be entered in the financial disclosure section of the submission system**." The Acknowledgment section should be at the end of the document. Please refer to https://journals.plos.org/plosone/s/submission-guidelines#loc-acknowledgments for more information.  Please submit your revised manuscript by Nov 14 2021 11:59PM. If you will need more time than this to complete your revisions, please reply to this message or contact the journal office at plosone@plos.org. Please include the following items when submitting your revised manuscript:A rebuttal letter that responds to each point raised by the academic editor and reviewer(s). You should upload this letter as a separate file labeled 'Response to Reviewers'.A marked-up copy of your manuscript that highlights changes made to the original version. You should upload this as a separate file labeled 'Revised Manuscript with Track Changes'.An unmarked version of your revised paper without tracked changes. You should upload this as a separate file labeled 'Manuscript'.If applicable, we recommend that you deposit your laboratory protocols in protocols.io to enhance the reproducibility of your results. Protocols.io assigns your protocol its own identifier (DOI) so that it can be cited independently in the future. For instructions see: https://journals.plos.org/plosone/s/submission-guidelines#loc-laboratory-protocols. Additionally, PLOS ONE offers an option for publishing peer-reviewed Lab Protocol articles, which describe protocols hosted on protocols.io. Read more information on sharing protocols at https://plos.org/protocols?utm_medium=editorial-email&utm_source=authorletters&utm_campaign=protocols.

We look forward to receiving your revised manuscript.

Kind regards,

Anna Manelis, Ph.D.

Academic Editor

PLOS ONE

Journal Requirements:

Reviewers' comments:

Reviewer's Responses to Questions

**Comments to the Author**

1. If the authors have adequately addressed your comments raised in a previous round of review and you feel that this manuscript is now acceptable for publication, you may indicate that here to bypass the “Comments to the Author” section, enter your conflict of interest statement in the “Confidential to Editor” section, and submit your "Accept" recommendation.

Reviewer #1: All comments have been addressed

Reviewer #2: All comments have been addressed

2. Is the manuscript technically sound, and do the data support the conclusions?

Reviewer #1: (No Response)

Reviewer #2: Yes

3. Has the statistical analysis been performed appropriately and rigorously? 

Reviewer #1: (No Response)

Reviewer #2: Yes

4. Have the authors made all data underlying the findings in their manuscript fully available?

Reviewer #1: (No Response)

Reviewer #2: Yes

5. Is the manuscript presented in an intelligible fashion and written in standard English?

Reviewer #1: (No Response)

Reviewer #2: Yes

6. Review Comments to the Author

Reviewer #1: (No Response)

Reviewer #2: Compared to the first draft, important changes have been made in the different sections of the paper. You tried to address the whole points raised by me.

1. As per the journal requirement “Acknowledgment” session is mandatory. So, try to include it after the conclusion session, and acknowledge people who were contributing for the development of this article.

7. PLOS authors have the option to publish the peer review history of their article (what does this mean?). If published, this will include your full peer review and any attached files.

Reviewer #1: No

Reviewer #2: No

---

## [Author Response · Author response to Decision Letter 1]

1 Oct 2021

Response to Reviewers

Manuscript ID: PONE-D-21-09795

Manuscript title: Brain Structures and Activity During a Working Memory Task Associated with Internet Addiction Tendency in Young Adults: A Large Sample Study

Journal: PLOS ONE

Dear editor and reviewers,

We sincerely thank the editor and reviewers for constructive and valuable comments, which were of great help in revising the manuscript. Accordingly, the revised manuscript has been systematically improved with new information and additional interpretations. Our responses to the editor comments (ARe), Journal Requirements (ARjr) and referee’s comments (ARr)are given below. Also, green text indicating changes has been added to the revised manuscript.

Editor comments

EC 1. "Do not include funding sources in the Acknowledgments or anywhere else in the manuscript file. Funding information should only be entered in the financial disclosure section of the submission system."

ARe 1. Thank you for your comment. We deleted the funding sources information in Acknowledgments and in the manuscript file. Funding information just entered in the financial disclosure section of the submission system.

Reviewer comment

RC 1. Compared to the first draft, important changes have been made in the different sections of the paper. You tried to address the whole points raised by me. 

As per the journal requirement “Acknowledgment” session is mandatory. So, try to include it after the conclusion session, and acknowledge people who were contributing for the development of this article.

ARr 1. Many thanks for your attention. We added “Acknowledgment” session in the end of the document as below:

Acknowledgment

We respectfully thank Yuki Yamada for operating the MRI scanner and Haruka Nouchi for acting as an examiner for psychological tests. We also thank study participants, the other examiners of psychological tests, and all of our colleagues at the Institute of Development, Aging and Cancer, Tohoku University, for their support.

Journal Requirements

JR 1: Please review your reference list to ensure that it is complete and correct. If you have cited papers that have been retracted, please include the rationale for doing so in the manuscript text, or remove these references and replace them with relevant current references. Any changes to the reference list should be mentioned in the rebuttal letter that accompanies your revised manuscript. If you need to cite a retracted article, indicate the article’s retracted status in the References list and also include a citation and full reference for the retraction notice.

ARjr: We reviewed our references carefully and have cited papers that have been retracted.

---

## [Editor Report · Decision Letter 2]

18 Oct 2021

Brain Structures and Activity During a Working Memory Task Associated with Internet Addiction Tendency in Young Adults: A Large Sample Study

PONE-D-21-09795R2

Dear Dr. Sadeghi,

We’re pleased to inform you that your manuscript has been judged scientifically suitable for publication and will be formally accepted for publication once it meets all outstanding technical requirements.

Kind regards,

Anna Manelis, Ph.D.

Academic Editor

PLOS ONE
---

## [Editor Report · Acceptance letter]

3 Nov 2021

PONE-D-21-09795R2 

Brain Structures and Activity During a Working Memory Task Associated with Internet Addiction Tendency in Young Adults: A Large Sample Study 

Dear Dr. Sadeghi:

I'm pleased to inform you that your manuscript has been deemed suitable for publication in PLOS ONE. Congratulations! Your manuscript is now with our production department. 

Kind regards, 

on behalf of

Dr. Anna Manelis 

Academic Editor

PLOS ONE